

# The median effective concentration of ropivacaine for ultrasound-guided anterior iliopsoas muscle space block in the elderly undergoing hip surgery: a dose-finding study

Peng Ma, Rui Zeng, Jiang Peng, Juan Zhu, Zhaojun Jing and Yu Han

Affiliated Hospital of Jiangsu University, Zhenjiang, China

## ABSTRACT

**Background:** In order to improve perioperative pain and reduce the adverse outcome of severe pain in elderly hip fractures, anterior iliopsoas muscle space block (AIMSB) can be used clinically to reduce pain. The aim of the study is to investigate the 50% effective concentration ($EC_{50}$) of ropivacaine for ultrasound-guided anterior iliopsoas space block in elderly with hip fracture.

**Methods:** A total of 27 patients were enrolled with aged ≥65 years, American society of Anesthesiologists (ASA) physical status classification II–III and undergoing Total Hip Arthroplasty (THA). We measured the $EC_{50}$ using Dixon's up-and-down method. Ultrasound-guided AIMSB was performed preoperatively with an initial concentration of 0.2% in the first patient. After a successful or unsuccessful postoperative analgesia, the concentration of local anesthetic was decreased or increased 0.05%, respectively in the next patient. The successful block effect was defined as no sensation to pinprick in the area with femoral nerve, obturator nerve, and lateral femoral cutaneous nerve in 30 min. Meanwhile, the $EC_{50}$ of ropivacaine was determined by using linear model, linear-logarithmic model, probit regression model, and centered isotonic regression.

**Results:** A total of 12 patients (48%) had a successful block. All patients with a successful block had a postoperative visual analog scale score of <4 in the 12 h. The estimated $EC_{50}$ values in linear model, linear-logarithmic model, probit regression model, and centered isotonic regression (a nonparametric method) were 0.268%, 0.259%, 0.277%, and 0.289%. The residual standard error of linear model was the smallest (0.1245).

**Conclusion:** The $EC_{50}$ of ropivacaine in anterior iliopsoas space block under ultrasound guidance is 0.259–0.289%.

# INTRODUCTION

The incidence of hip fractures in the elderly is increasing year by year. It is worth noting that Asia is often considered as a high-risk area for hip fractures (*Cooper et al., 2011*). According to the forecast model, the total number of hip fractures in Asia will increase

Corresponding author
Jiang Peng, doctorjp@163.com

from 1.12 million in 2018 to 2.56 million in 2050 (*Cheung et al., 2018*). In order to improve perioperative pain in elderly with hip fracture and reduce adverse outcomes due to severe pain, regional nerve blocks are often used as prophylactic analgesia clinically. Regional nerve blocks commonly used in the hip joint include: femoral nerve block (FNB), fascia iliac compartment block (FICB) (*Zhang & Ma, 2019*), pericapsular nerve group (PENG) (*Morrison et al., 2021*), erector spinae plane block (ESPB), anterior iliopsoas muscle space block (AIMSB) and lumbar plexus block, *etc.* Among them, *Dong et al. (2021)* showed that the anterior iliopsoas muscle space block had the same effect as the posterior lumbar plexus block on peri-operative analgesia for hip surgery. The femoral nerve descends between the psoas and iliac muscles, and descends into the femur through the fascia iliac *via* the psoas iliac muscle at the angle between the psoas iliac muscle. The obturator nerve travels posterior to the psoas major muscle and anterior to the obturator internal muscle, appearing at the medial border of the L5-level psoas major muscle. The lateral femoral cutaneous nerve runs obliquely and posteriorly and descendible through the fascia iliac through the anterior surface of the iliac muscle at the outer edge of the psoas major muscle obliquely. The space in the block area studied here is located at the level of the anterior superior iliac spine and formed between the psoas muscle and iliac muscle, and the femoral nerve, obturator nerve, and lateral femoral cutaneous nerve could be blocked by ultrasound-guided injection of local anesthetic into the iliopsoas space (*Laumonerie et al., 2021*).

Currently, the minimum effective concentration of ropivacaine for anterior iliopsoas space block has not been established. Due to exposure of peripheral nerves to the high concentration local anesthetics may contribute to neuronal damage (*Hogan, 2008*). The aim of study was to investigate the 50% effective concentration ($EC_{50}$) of ropivacaine for ultrasound-guided anterior iliopsoas space block in elderly with hip fracture o provide scientific reference for elderly medication.

## MATERIALS AND METHODS

This study was approved by the Ethics Committee of the Affiliated Hospital of Jiangsu University (KY2022H1209-2). This trial was registered on www.chictr.org.cn (ChiCTR2200066797; 17th of December 2022). The study is a part of the clinical trial registration protocol, and its primary purpose is to provide accurate and safe drug concentrations for the primary study in the protocol. Informed consent had been obtained in writing from patients prior to the conduct of the experiment. Recruitment of study was between December 20[th], 2022 and September 31[th], 2023 (Fig. 1).

The subjects were selected patients who underwent Total Hip Arthroplasty (THA) in the Affiliated Hospital of Jiangsu University, aged ≥65 years. Gender is not limited; ASA II-III; BMI 18.5–27.9 kg/m$^2$. If the patient has coagulation disorders; hypersensitivity to local anesthetics; infection at the puncture site; history of psychiatric or neurological illness and cannot communicate normally; severe systemic disease was excluded.

All patient routinely monitors pulse oxygen saturation, non-invasive blood pressure, electrocardiogram, establishes intravenous channels, and inhales oxygen by nasal cannula 4 L/min. The anterior iliopsoas muscle space block was performed 1 h before surgery. The
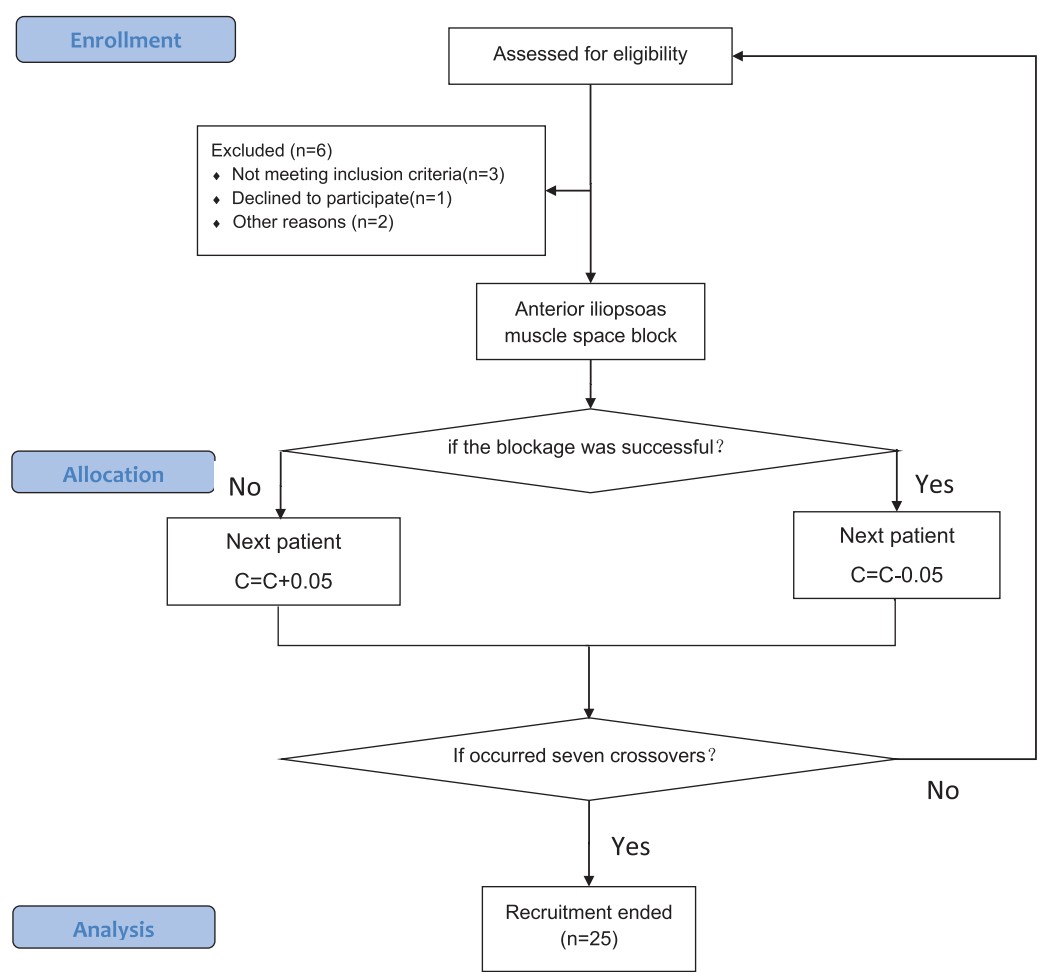

**Figure 1  Flow diagram of the CONSORT study.** The stages of the CONSORT study, beginning with the enrolment of the participants, followed by their allocation into intervention, and analysis.

patient was placed in the supine position and the puncture site disinfected with iodophor. THE 2–5 MHz convex array probe was placed on the medial side of the anterior superior iliac spine for cross-sectional scanning to Identify the psoas, iliac, and external iliac arteries (Fig. 2). The needle was injected into the iliopsoas space and the spread of local anesthetic was observed under ultrasound. A total of 25 ml of 0.2% Ropivacaine was administered in the first patient. Meanwhile, we conducted a pre-experiment and obtained that the 95% effective volume of 0.375% ropivacaine for ultrasound-guided anterior iliopsoas space block in elderly patients with hip fracture was 24.50 ml (95% CI [22.91~40.81] ml). Combined with the simplicity of clinical practice, the $EC_{50}$ of AIMSB was evaluated using 25 ml ropivacaine in this study.

By the Dixon and Mood's up-and-down study design (*Görges et al., 2017*), AIMSB concentration for subsequent patients was determined by success or failure of postoperative analgesia in the previous patient. Success or failure was judged by the presence or absence of a pinprick sensation to in the innervated area of the block within

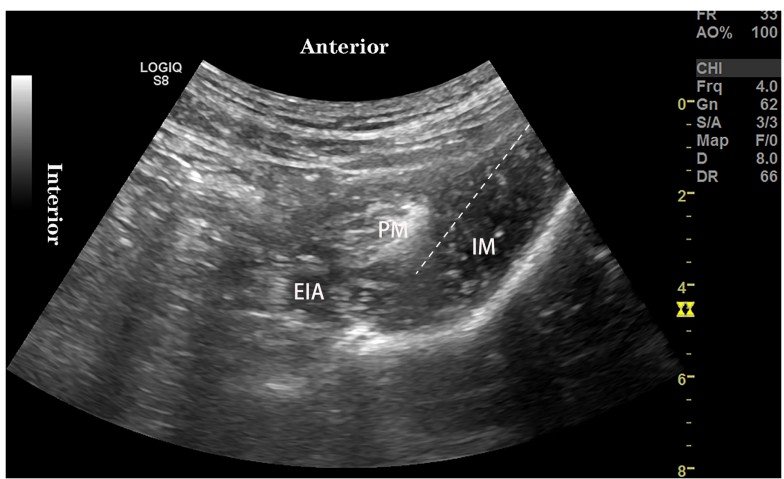

**Figure 2 Ultrasound-guided anterior iliopsoas muscle space block.** EIA, External iliac artery; PM, Psoas major muscle; IM, Iliacus muscle.               

30 min after the block. The concentration of ropivacaine was increased by 0.05% in case of failure and decreased by 0.05% in case of success (*Xu et al., 2022*).

The effects of sensory block were assessed every 10 min in 30 min post local anesthesia by an observer who was blinded to the concentration of local anesthetic injected, which performed preoperatively. Successful sensory block effect was defined as no sensation to pinprick in the area with femoral nerve, obturator nerve, and lateral femoral cutaneous nerve and graded as follows: 0 = normal sensation, 1 = blunted sensation, and 2 = no sensation (*Xu et al., 2022*). If total grade = 6, the block is considered successful. 30 min after the completion of AIMSB, the skin temperature in areas innervated by the femoral nerve, obturator nerve, and lateral femoral cutaneous nerve was monitored by an infrared thermal imager to assist in judging the effect of the block (*Asghar et al., 2014*). From the patient at a vertical distance of 2.0 m at room temperature (25.0 °C), an infrared thermal imager (model: HM-TPK20-3AQF/W) was used to detect the skin temperature below the umbilical plane and the increase in skin temperature on one side of blocking was considered as effective blocking. However, out of humanitarian and patient respect, we did not measure the patient's skin temperature continuously, only once 30 min after the blockade. Ultrasound-guided anterior iliopsoas space block was performed by an experienced physician who is well-versed in ultrasound-guided nerve block techniques, and was blinded to a physician who does not know the injection concentration within 30 min after the block.

After 30 min of effect evaluation, the patient's vital signs were confirmed to be stable, and the patient was transferred from the preparation room to the operating room for subarachnoid block. All patients underwent subarachnoid block injection of ropivacaine 12~15 mg. Palonosetron hydrochloride was applied to prevent postoperative nausea and vomiting. If hypotension occurs during surgery, ephedrine 6 mg or phenylephrine 40 μg was given intravenously. If VAS > 4 occurs postoperatively, parecoxib sodium 40 mg was given to remedy analgesia.

The visual analog scale (VAS) was used to record patients' pain. VAS of rest pain and movement-related pain was measured at 6, 12, and 24 h after the block time. The frequency of remedial analgesics was recorded. In addition, complications such as nausea and vomiting, nerve damage and infections related to anterior iliopsoas muscle space block during the perioperative period were recorded.

## Patient and public involvement

Patients and the public were not involved in the design and planning of the study. Patient and public representatives will be informed about the study. Patient and public involvement will be integrated during the study through working closely with patients who have underwent THA. A summary of the findings will be made available to the patient and public representatives.

## Statistical analysis

The exact sample size for Dixon's up-and-down method cannot be determined in advance. We stopped recruiting patients when seven crossovers (alternating between successful and unsuccessful blocks) occurred (*Zheng et al., 2022*).

IBM SPSS 27.0 software (IBM Corp., Armonk, NY, USA) and R statistical software version 4.3.2 (*R Core Team, 2024*) were used for statistical analysis. Count data are represented using the number of cases and percentage (%), normal distribution measures are expressed using mean ± standard deviation ($\bar{x} \pm s$), and nonnormal distribution measures are expressed as median (M) and interquartile range (IQR). The linear-logarithmic, linear and probit regression model were used to yield $EC_{50}$ and its 95% confidence interval (CI). A nonparametric model and the centered isotonic regression were also used to analyse a non-reducing dose and response relationship (*Pace & Stylianou, 2007*). The residual standard error was used to find the goodness of fit, which can analyse how well the data points fit the actual model. GraphPad Prism 9.5.1 software was used to draw sequential experimental plots. nonlinear fit curves and pain score *vs.* time-correlated line chart.

## RESULTS

Twenty-five patients were selected after seven crossovers up-down deflections (Fig. 3). The patient demographics characteristics (sex, age, BMI, ASA) and duration of surgery were well similar between successful and failed patients with block. There was no significant difference between both ($P > 0.05$) (Table 1).

Out of the total patients included in the study, 12 patients had a successful block. At the same time, an increase in skin temperature of the lower extremity on the block side was observed by thermography. All patients with a successful block had a postoperative visual analog scale score of <4 in the 12 h (Figs. 4A, 4B).

The estimated EC50 values in linear model, linear-logarithmic model, probit regression model, and centered isotonic regression (a nonparametric method) were 0.268%, 0.259%, 0.277%, and 0.289%, respectively (Fig. 5). The 95% confidence intervals for the three models (linear, linear-logarithmic and probit regression models) were 0.228%, 0.307%;

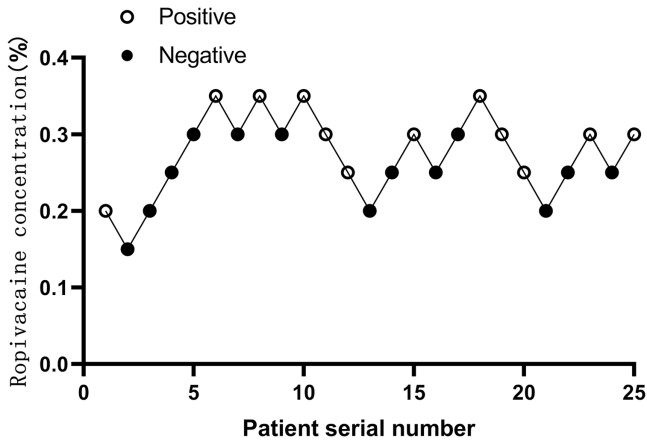

**Figure 3 Sequential block results of ultrasound-guided anterior iliopsoas muscle space block using 25 ml ropivacaine according to Dixon's up-and-down method.**

**Table 1 Characteristic of patients.**

| Characteristic | Positive group ($n = 12$) | Negative group ($n = 13$) | P value |
|---|---|---|---|
| Sex, $n$ (%) | | | 0.695 |
| Male | 6 (50) | 5 (38.5) | |
| Female | 6 (50) | 8 (61.5) | |
| Age (year), mean (SD) | 74.3 ± 5.47 | 76.77 ± 7.78 | 0.378 |
| BMI (kg/m$^2$) | 22.73 ± 2.22 | 22.96 ± 1.39 | 0.759 |
| ASA, $n$ (%) | | | 0.688 |
| II | 4 (33.3) | 6 (46.2) | |
| III | 8 (66.7) | 7 (53.8) | |
| Duration of surgery (min), mean (SD) | 76.75 ± 12.07 | 73.62 ± 17.07 | 0.604 |

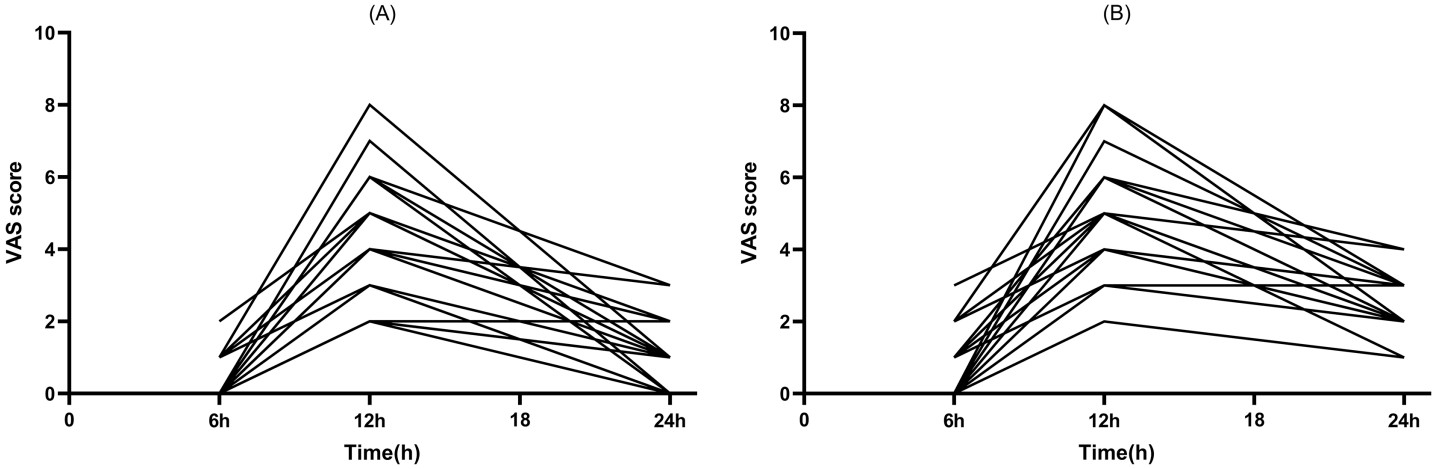

**Figure 4 Postoperative pain scores.** (A) Rest pain score 24 h after surgery. (B) Motor pain score 24 h after surgery.

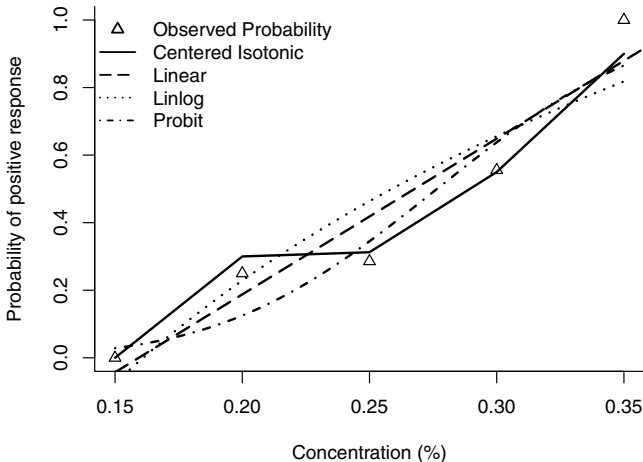

**Figure 5 Estimated ropivacaine concentration and anterior iliopsoas muscle space block relationship for a given dose level and probability of positive response.** Median estimators for each model are plotted.                                 

0.206%, 0.325%; and 0.240%, 0.313%, respectively. In these models, the residual standard error of linear model was the smallest (0.1245).

After surgery, four patients developed severe pinprick-like pain at the surgical site, and remedial analgesia was performed. One case occurred in the patients with successful block and three cases in the patients with failed block. There was no significant difference in the dose used for patients undergoing salvage analgesia between both.

None of the patients had complications such as bleeding infection, hematoma, and local anesthetic poisoning at the puncture site during nerve blockade; postoperative follow-up did not reveal nerve damage in the block area.

## DISCUSSION

The success of nerve blocks for analgesia is often judged clinically by hemodynamic stability and pain. It is well known that the concentration and volume of local anesthetic have a decisive effect on the effectiveness of nerve blocks. The degree of nerve damage caused by local anesthetics is concentration-dependent, and the higher the concentration, the more severe the nerve damage. Meanwhile, *Eledjam, Ripart & Viel (2001)* found that sensory and motor dissociative block was evident with low concentrations (0.2% and less) ropivacaine of block the femoral nerve. Therefore, we used an initial 0.2% concentration of ropivacaine to research the EC50 of AIMSB in this study. For the choice of drug volume aspect, we conducted a pre-experiment and obtained that the 95% effective volume of 0.375% ropivacaine for ultrasound-guided anterior iliopsoas space block in elderly patients with hip fracture was 24.50 ml (95% CI [22.91~40.81] ml). Compared with the supra-inguinal fascia iliac space block which need 40 ml 0.5% lidocaine to completely spread to the femoral nerve, obturator nerve, and lateral femoral cutaneous nerve (*Vermeylen et al., 2019*), the reason for this difference of volume with AIMSB may be due to the dense distribution of nerve locations in the iliopsoas space, and a satisfactory blockade effect can be achieved with a smaller volume of ropivacaine. Based on the ED95

study in our pre-experiments, we believe that the risk of local anesthetic poisoning of debilitating older patients with fractures can be reduce the risk in a certain extent by using a smaller volume of local anesthetic. Then, *Hu et al. (2023)* used 0.33% of 30 ml ropivacaine in the AIMSB group consuming significantly less morphine in the 24 h after surgery and the hospital stay than in the Sham group and had significantly lower pain scores during rest or exercise within 24 h of surgery. There were no significant differences in quadriceps muscle strength and postoperative complications between the two groups (*Hu et al., 2023*). That also demonstrated the safety and efficacy of AIMSB with lower concentrations and volumes of local anesthetics. However, this is also a limitation of this study, which did not corpse research to observe the extent of local anesthetic spread.

*Girón-Arango et al. (2018)* proposed PENG block which is an injection of local anesthetic around the anterior capsular of the hip joint, it is effective in blocking the sensory nerve of the hip joint. The target location of PENG block is at the myofascial plane between anterior to the iliopsoas tendon and posterior pubic ramus. Similar to AIMSB, PENG equally retains quadriceps muscle strength. In a review of studies based on the widespread use of PENG block in postoperative hip pain management, several RCTs comparing the effects of different nerve block methods on quadriceps muscle strength or postoperative rehabilitation were analyzed in detail, and compared with other nerve blocks, PENG block reduced the incidence of quadriceps weakness, significantly shortened bed rest, and extended the distance to first walk. Other hand, due to the proximity of the puncture site to the surgical site, improper manipulation may lead to joint infection. The route of AIMSB is far away from the internal organs, abdominal cavity, and blood vessels, and can achieve the ideal analgesic effect while reducing the risk of damage to internal organs and blood vessels.

The anesthesia used to meet surgical needs in the study protocol was a single subarachnoid block with ropivacaine, which typically lasted 2 to 4 h (*Kumar & Seet, 2023*). To reduce the effect of subarachnoid block on pain perception assessment, rest pain scores and motor pain scores were assessed at 6, 12, and 24 h postoperatively. In this study, we found varying degrees of increase in pain scores at rest and exercise at 12 h after surgery, but no significant increase in pain scores at 6 h after surgery. The main cause may be due to an anterior iliopsoas space block time of less than 12 h.

Some studies found that regional block anesthesia not only blocks sensory nerves and motor nerves, but also had a blocking effect on sympathetic nerves, which blocks the expansion of blood vessels in sympathetic innervation, increases blood flow, and increases skin temperature (*Van Haren, Kadic & Driessen, 2013*). In this study, the increase in skin temperature in blockade innervation was judged by infrared thermal imager, which more objectively reflected the success of the blocking effect and the range of blockade, especially in older patients who are unable to accurately describe feelings. In this study, the patients with successful block showed significant differences in thermal imaging images in the area of blockade-femoral nerve, obturator nerve, and lateral femoral cutaneous nerve innervation compared with the skin temperature of the contralateral limb. However, out of humanitarian and patient respect, this study only performed thermal imaging 30 min after blockade. Even though studies have demonstrated a significant temperature increase

15 min after blockade (*Yoshimura et al., 2021*), it is not possible to accurately describe the success time of the blockade in the innervated region.

The Dixon up-and down method allowed the determination of an $EC_{50}$ for a clinical variable in the small sample size study and widely used in the field of anesthesia (*Dixon, 1991*). The advantage of this approach is that it requires a small number of patients, which can save a large sample size and reflect the situation at any point in the dose-response curve. Most studies recognized that when there are seven effective-inflection points in the trial sequence, the sample size that stops recruiting can reflect the effective amount of the drug being measured. But the Dixon up-and down method is inaccurate when small or large percentage points are calculated, such as $EC_{95}$. Our simulations result of 25 small samples maybe reliable for $EC_{50}$ but significantly less accurate for $EC_{95}$.

## CONCLUSIONS

Sum up, the $EC_{50}$ of ropivacaine in anterior iliopsoas space block under ultrasound guidance is 0.259–0.289%, and linear model best matched the data. At the same time, this method of anesthesia is safe and effective for elderly undergoing THA surgery.

### Funding

This research was funded by the Science and Technology Planning Social Development Project of Zhenjiang City (SH2022083). The funders had no role in study design, data collection and analysis, decision to publish, or preparation of the manuscript.

### Grant Disclosures

The following grant information was disclosed by the authors:
Science and Technology Planning Social Development Project of Zhenjiang City: SH2022083.

### Competing Interests

The authors declare that they have no competing interests.

### Author Contributions

- Peng Ma conceived and designed the experiments, performed the experiments, authored or reviewed drafts of the article, and approved the final draft.
- Rui Zeng performed the experiments, analyzed the data, prepared figures and/or tables, and approved the final draft.
- Jiang Peng conceived and designed the experiments, authored or reviewed drafts of the article, and approved the final draft.
- Juan Zhu performed the experiments, authored or reviewed drafts of the article, and approved the final draft.
- Zhaojun Jing analyzed the data, prepared figures and/or tables, and approved the final draft.

- Yu Han conceived and designed the experiments, authored or reviewed drafts of the article, and approved the final draft.

## Human Ethics

The following information was supplied relating to ethical approvals (*i.e.*, approving body and any reference numbers):

Ethics Committee of the Affiliated Hospital of Jiangsu University.

## Clinical Trial Ethics

The following information was supplied relating to ethical approvals (*i.e.*, approving body and any reference numbers):

This study was approved by the Ethics Committee of the Affiliated Hospital of Jiangsu University (KY2022H1209-2).

## Data Availability

The raw measurements are available in the Supplemental Files 1.

## Clinical Trial Registration

The following information was supplied regarding Clinical Trial registration:

ChiCTR2200066797.

## Supplemental Information

Supplemental information for this article can be found online at http://dx.doi.org/10.7717/peerj.17970#supplemental-information.

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
