# Peer review of "The median effective concentration of ropivacaine for ultrasound-guided anterior iliopsoas muscle space block in the elderly undergoing hip surgery: a dose-finding study"

_PeerJ, doi:10.7717/peerj.17970_

## Round 0.1 · original submission · Major Revisions

Dear Dr. Peng,

If you feel you can revise your manuscript according to the reviewers' comments, please revise your manuscript and submit it. Please also send us your written responses to each of the reviewers' comments.

Yours,

Yoshi

Prof. Yoshinori Marunaka, M.D., Ph.D.

Reviewer 1 ·

Basic reporting

I have reviewed the abstract, introduction, methods and materials, results, statistics, and discussion. I have also checked the references, and all appear relatively current and appropriate. Finally, I have also reviewed the figures, tables, and legends.

I appreciate the clarity and complexity of the presentation. The figures and tables are representative. However, I have a few suggestions aiming to improve the manuscript quality:
- In the Introduction:
o Why is Asia considered a high-risk area for hip fractures? Please provide reference
- In the Methods Section- several discrepancies appear between your trial and the trial registered in the Chinese Clinical Trial Registry.
o Different title
o Different Objectives of the study
o Different methods
o Different outcomes
o Different groups
- You have registered for a different study or provided an incorrect trial registry number. Please explain
- In the methods section, please explain when the AILSB was performed and when the subarachnoid block was performed.
o Did you perform the peripheral nerve block before spinal anesthesia? How long did you observe the evaluation of the block?
o Why did you choose 25ml of Ropivacaine? Why not 20 or 15?
- Discussion
o Please explain how the AILSB differs from the PENG block.

Experimental design

no comment

Validity of the findings

no comment

Additional comments

no comment

Reviewer 2 ·

Basic reporting

Some sentence are still not unclear and ambiguous. Please check your grammar and common related PNB words.

Experimental design

Please tell more information in introduction and methods parts.
The final successful PNB number of patients and adequacy of sample size.

Validity of the findings

The clinical implication may be added in the discussion part.

Additional comments

I attach my review file.

Annotated reviews are not available for download in order to protect the identity of reviewers who chose to remain anonymous.

Reviewer 3 ·

Basic reporting

Dear Authors,
Thank you very much for submitting your manuscript, "The Median Elective Concentration of Ropivacaine for Ultrasound-Guided Anterior Iliopsoas Muscle Space Block in the Elderly Undergoing Hip Surgery: A Dose-Finding Study," to PeerJ. I appreciate the considerable effort and dedication invested in this work and congratulate you on it. However, I believe the manuscript has several critical deficiencies that need to be addressed.
General Comments:
The manuscript requires a coherent thread. The stated objectives must be related to a hypothesis, and the experimental design must be aligned to prove or refute the hypothesis(es) posed. The materials and methods section should provide the necessary information to reproduce the study in other centers and with other patients. The details presented in the current manuscript do not meet these requirements.
A chronological sequence of the procedure helps to follow the story and makes the procedure more understandable. This sequence needs to be included in this presentation.
The introduction should prepare the reader to understand the reason for the study. In this case, procedures are carried out that cannot be understood. For example, what is the point of presenting thermography results?
Finally, the discussion section should discuss the results obtained and compare them with similar or dissimilar results from other studies. For example, almost the entire discussion is used to debate issues unrelated to the research presented. I invite the authors to rearrange the text, creating a concise, sequential, and straightforward report focusing exclusively on the study, objectives, and hypotheses.

Experimental design

Specific Comments:
Abstract: Ensure consistency between the text and the abstract, such as specifying ASA I-III or ASA II-III.
Keywords: Keywords should not be words included in the title. Please apply general rules for keywords, and I suggest using MeSH terms (http://www.ncbi.nlm.nih.gov/entrez/query.fcgi?db=mesh).
Line 77: State the method used to assess pain during recovery.
Line 78: Indicate the cutoff for determining success/failure.

Validity of the findings

Line 80: Clarify that this evaluation was performed during pre-op.
Line 82: Pinprick evaluates skin dermatomes, which are innervated by nerve fibres different from those that innervate the hip joint. What was the rationale for including this step in the study? Is the EC50 for the cutaneous and articular branches comparable? What about infrared thermal images?
Line 89: Define how many successful block definitions were used.
Line 93: The evaluation timing of the block could be more precise. A detailed description of the procedure should be provided. Please reorganize this section for clarity.
Line 140: Presenting successful and failed outcomes as "groups" is confusing since no groups were previously defined. Please amend.
Lines 147-157: This information needs to be revised and belongs in the discussion section. Please remove.

Additional comments

I understand that my review is critical and severe. I will guide you toward achieving the necessary level to publish your manuscript in this prestigious journal. I hope my comments do not discourage you. I recognize the many hours of work and resources behind this attempt and commend you for your dedication.
I hope my comments and challenges guide you well for future studies.
Best regards,

---

## Round 0.2 · accepted · Accept

Congratulations again, and thank you for your submission.

Best regards,
Yoshi
Prof. Yoshinori Marunaka, M.D., Ph.D.
Academic Editor
PeerJ Life & Environment